# Iontronic click-to-release enables electrically controlled delivery of drugs and biomolecules beyond charge and size limitations

Sebastian Hecko [1,6], Marle E. J. Vleugels [2,6], Christian Bayer [3,6], Donghak Byun [2], Moa E. Hörberg [2], Nikolaus Poremba [1], Rassen Boukraa [2], Patrick Keppel [1], Andreas Löffler [1], Walter Kuba [1], Helena Saarela Unemo [2], Iwona Bernacka-Wojcik [2], Theresia Arbring Sjöström [2], Magnus Berggren [2,4], Daniel T. Simon [2], Rainer Schindl [3], Linda Waldherr [3,5] ✉, Hannes Mikula [1] ✉ & Johannes Bintinger [1,2] ✉

Dynamic and programmable control of therapeutic delivery is a long-standing goal in medicine. Iontronic devices offer precise electronic control over the dosage of bioactive molecules, yet their use has been confined to charged, low-molecular-weight compounds that are electrochemically stable during transport. Here, we present a hybrid delivery platform that integrates iontronic transport with bioorthogonal click-to-release chemistry. In this system, iontronic pumps electrophoretically deliver charged tetrazines as molecular scissors that selectively react with immobilized *trans*-cyclooctene (TCO)-linked payloads, enabling on-demand bioorthogonal cleavage of the TCO linker and controlled payload release. This approach retains the electronic precision of iontronics while overcoming molecular size, charge, and stability constraints. We demonstrate tunable tetrazine delivery over several days and electronically controlled release of immobilized payloads from small bioactive molecules, such as the antimitotic agent CA4, to the large protein bovine serum albumin. Hence, by integrating bioorthogonal click-to-release strategies, iontronic delivery is extended to biologically relevant macromolecules, providing a foundation for advanced programmable electroceutical devices.

Modern medicine increasingly recognizes that therapeutic efficacy depends not only on what drug is delivered, but also where, when, and how it is administered. The ideal therapeutic platform would achieve precise spatiotemporal control over drug dosing, dynamically adjusting to physiological needs, minimizing systemic exposure, and enabling personalized treatment profiles[1,2]. Particularly for applications inside the human body, locally restricted release of drugs using different types of hydrogels[3], nanocarriers[4], and stents[5] has gained

[1]Institute of Applied Synthetic Chemistry, TU Wien, Vienna, Austria. [2]Laboratory of Organic Electronics, Department of Science and Technology, Linköping University, Norrköping, Sweden. [3]Gottfried Schatz Research Center - Medical Physics and Biophysics, Medical University of Graz, Graz, Austria. [4]Wallenberg Initiative Materials Science for Sustainability, Department of Science and Technology, Linköping University, Norrköping, Sweden. [5]BioTechMed-Graz, Graz, Austria. [6]These authors contributed equally: Sebastian Hecko, Marle E. J. Vleugels, Christian Bayer. ✉e-mail: linda.waldherr@medunigraz.at; hannes.mikula@tuwien.ac.at; johannes.bintinger@tuwien.ac.at

considerable attention in the last years, showcasing an increased therapeutic effect while minimizing adverse systemic effects. However, current therapeutic implants lack real-time adaptability and offer limited control over dose timing[6–9]. Put simply, once the drug release has been initiated, it cannot be turned off or adjusted. Many of these systems also suffer from the burst release effect[10,11], characterized by an initial rapid release of a large drug fraction before switching to a constant passive release profile (red trace in Fig. 1). The inability to reprogram or modulate therapy by adjusting drug delivery profiles in situ compromises their therapeutic safety and limits their clinical impact[12]. Implantable devices and osmotic pumps have been successfully designed to deliver drugs locally[4,13–20], but these technologies typically rely on pre-programmed release or external triggers. Such triggers can be chemical (e.g., pH changes/hydrolysis[21,22], enzyme-mediated cleavage[23,24], oxidative stress/ROS[25–27]) or physical (e.g., magnetism[28–30], temperature[31], ultrasound[32,33] or light[34–36]). The latter enable remote activation, converting externally applied energy into localized effects such as heating, mechanical stress, or photon-induced activation at the target site. Depending on the trigger platform used, this may require implantable release reservoirs or local administration of stimulus-responsive carriers (e.g., nanoparticles) that can be subsequently activated in situ. As activation relies on energy propagation through biological tissue, achieving precise spatial targeting and accurate dose control remains challenging. In addition, such energy transfer can adversely affect surrounding tissue and, for light-based systems, typically requires optical access to the target region. Iontronic pumps (IP) offer an alternative to several of these limitations by converting electrical signals directly into molecular transport, providing tunable delivery rates with high spatiotemporal control. IPs consist of (i) a source (drug) reservoir, (ii) an ion exchange membrane (IEM) forming the "channel" which provides selective transport of charged species from the reservoir to the target region, and (iii) electrodes in reservoir and target regions[37,38]. By applying a defined potential difference between source and target electrodes, species with opposite charge to the fixed charges in the IEM are electrophoretically transported ("pumped") at a specific rate across the IEM into the target site without any liquid flow or the need for mechanical components. This enables the system to translate electrical signals into precise and customizable biochemical concentration-time profiles. IPs have been successfully used to control the delivery of small molecules such as neurotransmitters[39–41], plant hormones[42,43], modulators of inflammation[44,45], and chemotherapeutics[46–48]. However, to ensure effective charge selectivity through the IEM (and thereby dose precision), the effective pore size in the hydrated IEM material limits IP-delivery to charged (ionic) compounds with low molecular weight (typically $\lesssim$600 Da), which are (electro)-chemically stable. Consequently, these restrictions severely limit the scope of therapeutics that are compatible with iontronic transport[49].

To overcome these limitations, we combined iontronic delivery with subsequent click-to-release (C2R) chemistry to effectively decouple the transport mechanism from payload liberation or drug activation. C2R reactions are a class of bioorthogonal reactions in which a chemical trigger selectively reacts with a latent payload-bearing molecule, resulting in the initial formation of a covalent intermediate "click" product that undergoes rapid internal rearrangement "to release" the payload[50]. This concept has previously been applied for targeted drug activation in vivo, including in hydrogel depot systems and antibody-drug conjugates[51–54]. Although these approaches offer localized therapeutic action, they depend on systemic dosing of prodrugs and afford only limited control over the timing of activation. In contrast, the approach presented here demonstrates electronically controlled activation and enables tunable dosing. To implement this concept, we employed the click-triggered bond-cleavage chemistry of *trans*-cyclooctenes (TCO) and tetrazines (Tz), known for its exceptional kinetics and proven compatibility with biological systems[55]. In particular, aminoethyl-functionalized Tz were shown to be efficient reagents[56] to accelerate the click-to-release of cleavable TCO and, due to their permanent charge under physiological conditions, compatible with IPs. We therefore used symmetric

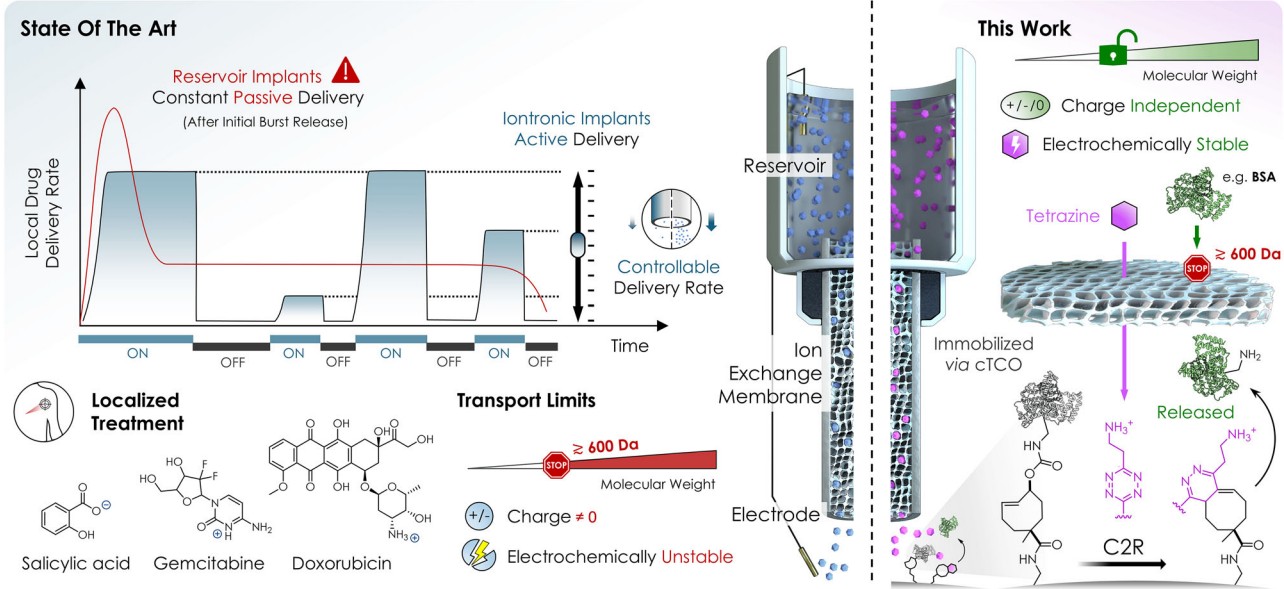

**Fig. 1 | Schematic illustration of an iontronic click-to-release platform enabling prodrug activation beyond conventional iontronic delivery limits.** Left—State of the art: Local reservoir-based implants for drug delivery purposes typically exhibit an initial burst followed by constant passive release that cannot be modulated during treatment. In contrast, iontronic delivery *via* ion exchange membranes (IEMs) enables active and temporally controlled dosing but is limited to small ($\lesssim$600 Da), charged, and electrochemically stable compounds (e.g., salicylic acid[45],

gemcitabine[46], or doxorubicin[48]). **Right**—This work: The click-to-release (C2R) strategy overcomes these limitations by transporting charged aminoethyl tetrazine through the IEM to react with *trans*-cyclooctene (TCO)-immobilized prodrugs, enabling localized release of unmodified drugs independent of molecular weight, charge, or electrochemical stability. Certain graphical elements in this figure were created in BioRender. Hecko, S. (2026) https://www.BioRender.com/z8yr1eu.

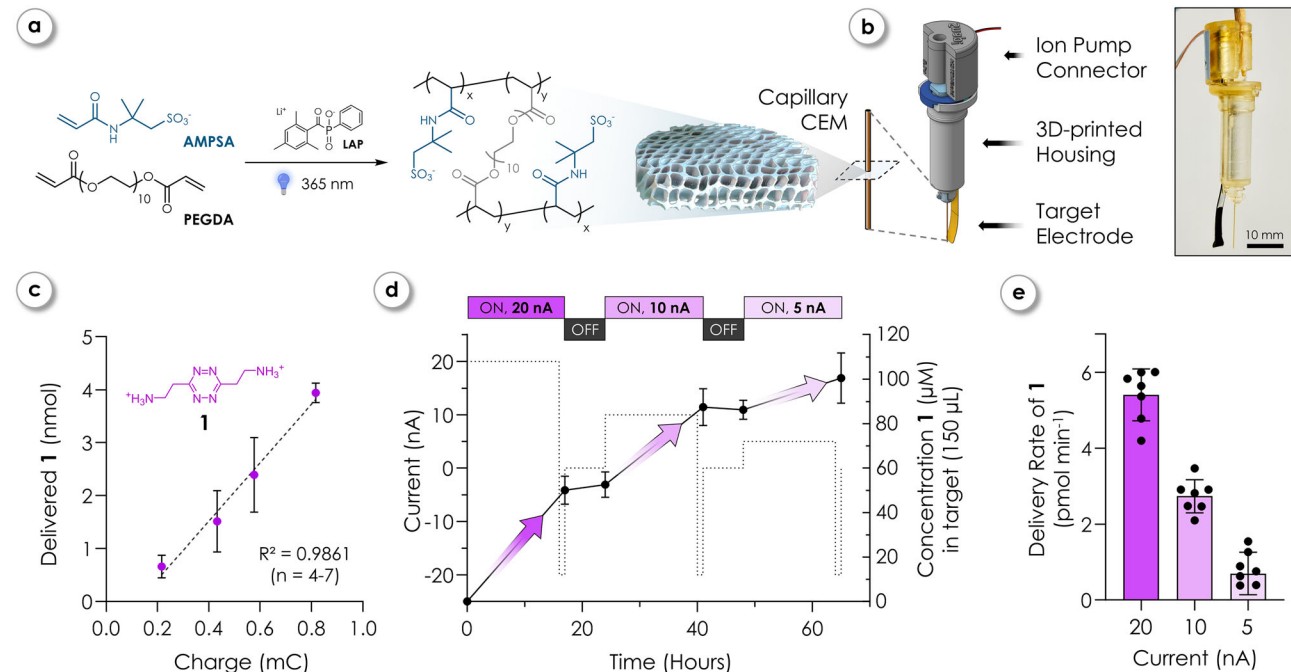

**Fig. 2 | Iontronic transport and programmable delivery of charged tetrazine through a CEM-based IP. a** Formation of the negatively charged cation exchange membrane (CEM) *via* UV-initiated crosslinking of AMPSA and PEGDA using LAP as a photoinitiator. **b** Schematic of IP incorporating the CEM within the polyimide capillary mounted in a custom 3D-printed housing; photograph shows assembled device. **c** Delivered amount of aminoethyl tetrazine (**1**) as a function of total applied charge (current × time). Data are shown as mean ± SD; $n = 4-7$ independent experiments per data point. Data were fitted by linear regression ($R^2 = 0.9861$). **d** Temporal control of delivery using alternating "ON" and "OFF" phases: ON phases involved application of +20, +10, or +5 nA for 16 h, each followed by a brief −20 nA

pulse (1 h); OFF phases consisted of 0 nA for 7 h. Data are shown as mean ± SD; $n = 7$ (+ 20 nA), $n = 6$ (+10 nA), and $n = 8$ (+ 5 nA) independent experiments. **e** Delivery rates of **1** (pmol min⁻¹) corresponding to each "ON" phase in (**d**). Rates were calculated from the same datasets shown in (**d**). Bars represent mean ± SD. All data in (**c**–**e**) were quantified *via* the C2R using fluorogenic rTCO-DMEDA-4MU (**2**) for selective detection of pumped Tz **1** *via* **4MU** release. Delivered amounts of **1**, the applied charge, and the delivery rates were calculated using the Eqs. (1–4) provided in the Methods section. Certain graphical elements in (**a**) were created in BioRender. Hecko, S. (2026) https://www.BioRender.com/z8yr1eu.

aminoethyl tetrazine (**1**, Fig. 2c) as chemical trigger, which was sufficiently small to be transported through the IEM (< 200 Da) while offering the required high solubility and reactivity properties. Payloads were immobilized onto bead-scaffolds, which essentially served as on-demand activatable drug reservoirs *via* click-cleavable TCO linkers (cTCO, Fig. 1). This allowed for payload modifications without being constrained by the charge or molecular weight limitations inherent to iontronic delivery. We initially demonstrated the controlled, programmable iontronic transport of Tz **1** and later showcased the controlled release of payloads using both the small-molecule chemotherapeutic combretastatin A-4 (**CA4**, 316 Da) and bovine serum albumin (**BSA**, 66 kDa) from magnetic beads.

## Results

### Controlled iontronic tetrazine delivery

Building on designs and results of controlled compound delivery using established capillary IPs[43,57], we aimed to establish reliable Tz **1** delivery. We initially confirmed that aminoethyl tetrazine (**1**) remained intact under iontronic delivery conditions by cyclic voltammetry (Supplementary Information 9.2). The polymeric matrix enabling iontronic delivery of positively charged **1** was formed as negatively charged sulfonated cation-exchange membrane (CEM) within polyimide-coated fused silica capillaries (Fig. 2a) by UV-polymerizing 2-acrylamido-2-methylpropane sulfonic acid (AMPSA), poly(ethylene glycol) diacrylate (PEGDA, Mn 575), and photoinitiator (lithium phenyl-2,4,6-trimethylbenzoylphosphinate, LAP)[58]. All capillary IPs were fabricated based on previously reported procedures with improved custom-made Eppendorf or well-plate compatible 3D-printed housings (Figs. 2b and 4b and Supplementary Information 4)[57].

IP performance was initially tested with the small cation K⁺ (0.1 M KCl) in the source before switching to Tz loading and subsequent delivery (Supplementary Information 5). The representative chrono-potentiometry shows a stable low-voltage (~1 V) operation when a constant current (nanoamperes) was applied to the system (Supplementary Fig. S6). The amount of delivered **1** was determined using a fluorogenic assay *via* the C2R of 4-methylumbelliferone (**4MU**) from click-cleavable TCO-tag rTCO-DMEDA-4MU (**2**) and enabled quantification of **1** down to micromolar concentrations. An increase in applied charge (current × time) could be directly correlated to the amount of delivered **1**, demonstrating full control over the iontronic transport (Fig. 2c) with an electrochemical transport efficiency of approximately 68% at +20 nA.

To validate the programmability and reliability of the transport, we operated devices for 2.5 days with alternating "ON" and "OFF" phases (16 h vs. 7 h, Fig. 2d), showcasing the stability, necessary for combining iontronic transport with the activation of (highly) potent drugs. Decreasing the applied current from +20 to +5 nA was shown to follow a linear trend. Delivery rates were 5.4 ± 0.7, 2.7 ± 0.4, and 0.7 ± 0.6 pmol min⁻¹ for a constant applied current of +20, +10, and +5 nA, respectively (Fig. 2e and Supplementary Fig. S9). Short reversal of the current (−20 nA for 1 h) followed by removal of the active bias was shown to be enough to effectively halt passive diffusive processes through the membrane (Supplementary Fig. S8). Given an example target volume of 10 mL, these rates would correspond to an increase in the concentration of **1** on the order of ~0.5 nM min⁻¹ during +20 nA active delivery. Considering the subsequent C2R kinetics (Supplementary Information 3.2), this release rate would enable efficient local release of highly potent drugs in the nM to μM concentration range over therapeutically relevant timescales of minutes to hours.

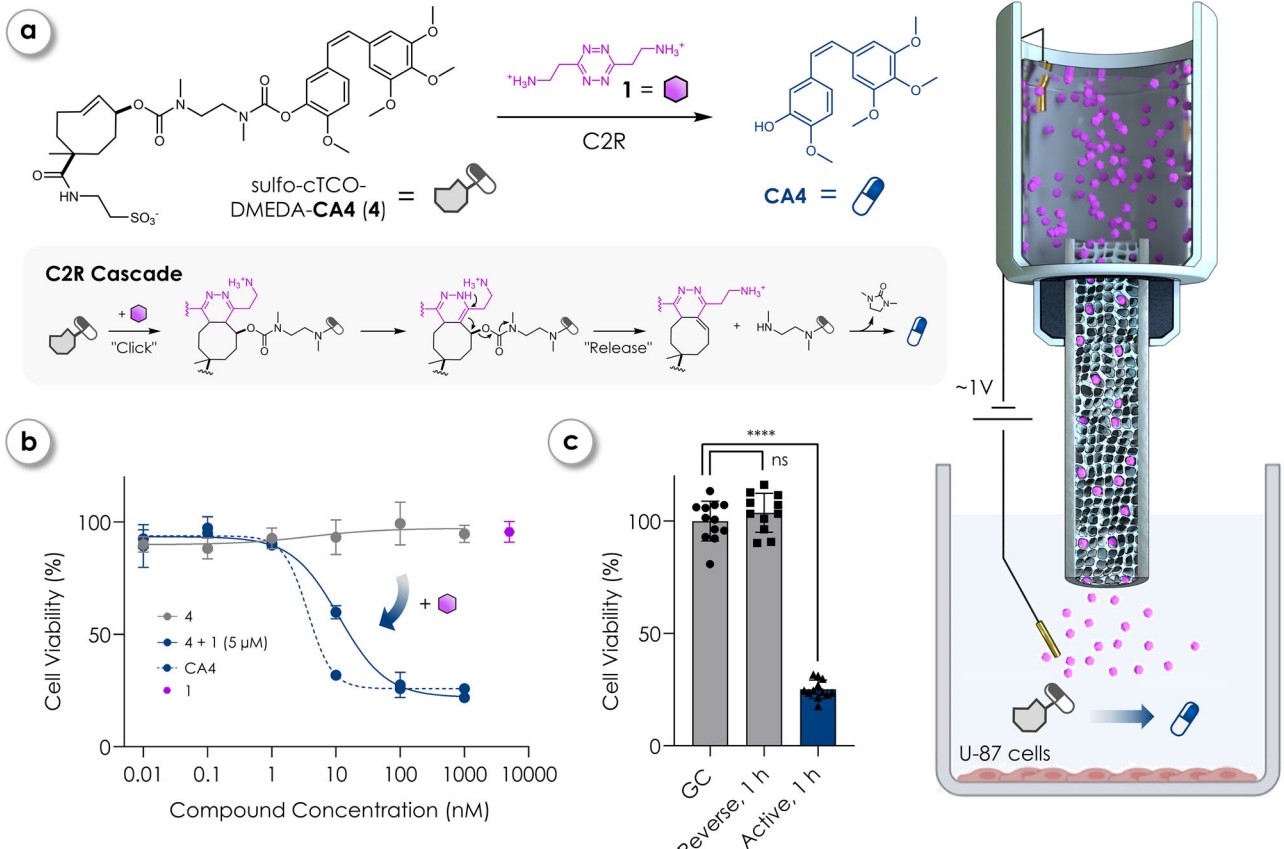

**Fig. 3 | Iontronic Tz delivery and prodrug activation via click-to-release enables localized cytotoxicity in vitro. a** Schematic overview of the bioorthogonal click-to-release (C2R) cascade of CA4-prodrug: Aminoethyl tetrazine (Tz **1**) triggers the release of the cytotoxic agent combretastatin A-4 (**CA4**) from sulfo-cTCO-DMEDA-CA4 (**4**) *via* a tetrazine-triggered elimination from TCO caged payload and self-immolation. **b** Dose-dependent cell viability of human glioblastoma cell line U-87 after 72 h incubation with the key components of the C2R reaction. Data represent mean ± SD of *n* = 3 independent biological replicates. Curves were fit using four-parameter logistic regression. Prodrug **4** (0.01–1000 nM, grey curve); parent **CA4**

(0.01–1000 nM, blue curve); released **CA4** (0.01–1000 nM **4** + 5 μM **1**, dashed blue curve); Tz **1** (5 μM, purple data point). **c** Cell viability of U-87 cells after operating iontronic devices delivering **1** for 1 h at +20 nA (Active, *n* = 9) and at −20 nA (Reverse, *n* = 11). Points represent individual biological replicates; bars indicate mean ± SD. GC growth control (untreated, *n* = 12). Statistical analysis was done *via* a Brown–Forsythe and Welch ANOVA tests (one-way) and Dunnett's T3 multiple comparisons test. α = 0.05. *p*-values: GC vs. Reverse, 1 h (ns) = 0.5339; GC vs. Active, 1 h (****) = <0.0001. Certain graphical elements in (**a**) were created in BioRender. Hecko, S. (2026) https://www.BioRender.com/z8yr1eu.

## CA4 prodrug activation

To further demonstrate the controllability and safety of the iontronic Tz pumping for drug activation, we tested the release of the highly potent anti-mitotic and anti-angiogenic agent combretastatin A-4 (**CA4**, IC$_{50}$ ~1 nM for HT1080 human fibrosarcoma cells)[59]. We used click-cleavable TCO-linker (cTCO)[52], already used as caging moiety for **CA4** and evaluated its C2R kinetic parameters with Tz **1**. Second-order rate constant $k_2$ was determined using stopped-flow kinetics between cTCO-(PEG$_2$-OH)$_2$ (**3**) to **1** to be 37 M$^{-1}$ s$^{-1}$ (Supplementary Fig. S1). The release process was studied by HPLC analysis using water soluble sulfo-cTCO-DMEDA-CA4 (**4**) prodrug with **1** (Fig. 3a) and was shown to be highly efficient, achieving near complete (> 94%) **CA4** release within 3 h at micromolar concentrations (Supplementary Information 3.2). To confirm that release also occurs under iontronic conditions, we performed HPLC analysis during electronic delivery of **1** into a reservoir of prodrug **4**. **CA4** release increased proportionally with pumping duration (3–9 h), whereas K$^+$ delivery did not induce any detectable release, confirming mechanistic fidelity and prodrug stability under iontronic operation (Supplementary Information 6.1). Initial cell viability experiments with soluble CA4-prodrug **4** confirmed that **4** is non-toxic for the human glioblastoma cell line U-87 at nanomolar concentrations, but that a combination treatment of increasing concentrations of the prodrug **4** + 5 μM **1** showed comparable cell toxicity to parent **CA4** (Fig. 3b) with a minimal cell viability of ~25%. Increasing **CA4**

concentration beyond this point did not lead to any additional decrease in viability, corresponding to the maximal toxic effect of **CA4** in this system (Supplementary Information 6.2). Neither the TCO-bound prodrug **4** (up to 1 μM) concentration, nor Tz **1** (5 μM) affected the cells (99% cell viability). Both the released (**4** co-incubated with 5 μM **1**) and parent **CA4** showed similarly reduced cell viability at 100 nM concentrations of 28% and 26%. In other words, the non-released prodrug **4** is effectively non-toxic until the release of **CA4** is triggered by **1**.

We therefore installed iontronic devices delivering **1** over cells in 100 nM **4** and operated the devices in forward (+) and reverse (−) constant current mode for 1 h at ± 20 nA (Fig. 3c, potential traces shown in Supplementary Fig. S12), followed by removal of the pumps and subsequent 72 h incubation. One device and its corresponding cell viability data was excluded due to reaching the maximum voltage of 10 V, indicating non-functional Tz delivery (Supplementary Fig. S13). Based on the established delivery rate of **1** at +20 nA, we estimated that active pumping for just 30 s would suffice to deliver ~10 nM of Tz **1** into 250 μL, thereby releasing a biologically relevant amount of **CA4**. While active delivery significantly lowered cell viability to 25%, correlating with the maximal toxic effect of **CA4** in the U-87 cell line, the reverse operation effectively halted transport of **1** across the CEM and thus did not influence cell viability, exemplifying the high level of control achieved with iontronic delivery of the C2R trigger.

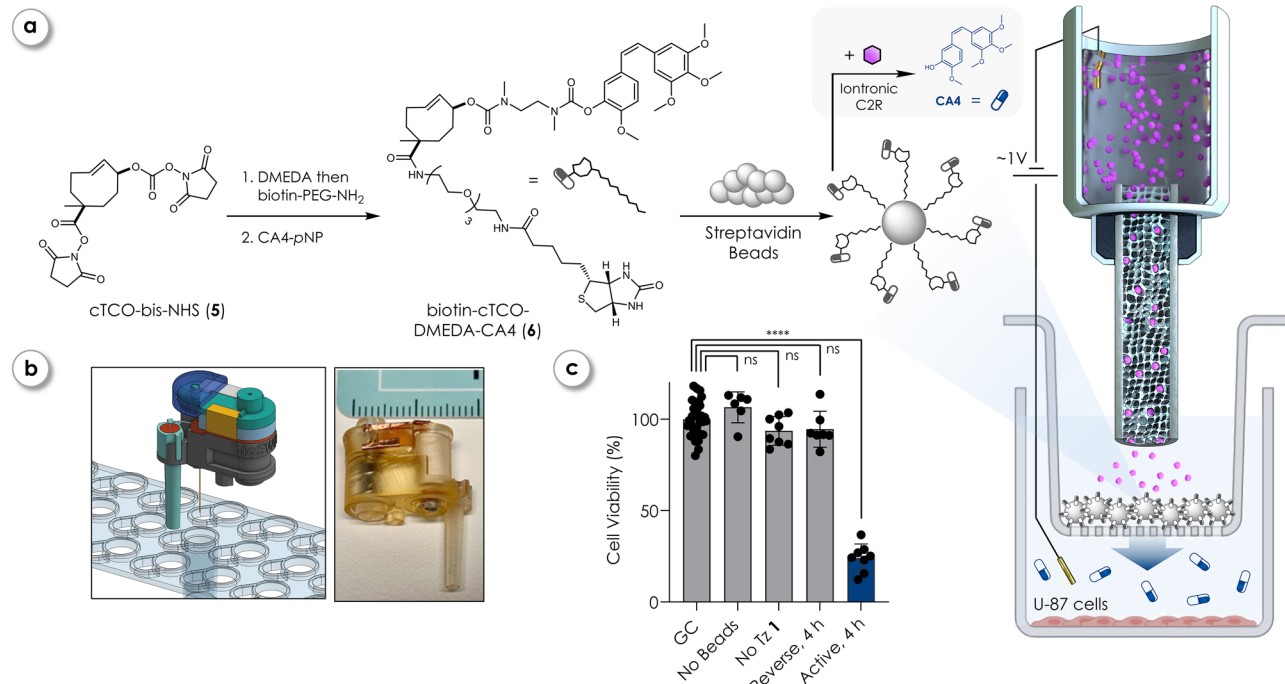

**Fig. 4 | Iontronic Tz delivery and click-to-release of bead-immobilized CA4-prodrug. a** Schematic of the synthesis and immobilization strategy for biotinylated CA4-prodrug (**6**), synthesized *via* sequential decoration of cTCO-bis-NHS (**5**). The biotin handle allows immobilization onto streptavidin-coated magnetic beads, forming a localized, iontronically activatable prodrug reservoir. IPs were positioned above the beads in a transwell insert containing U-87 cells in the lower chamber. Grey pills indicate inactive (bound) drug, blue (colored) pills indicate active drug. **b** 3D-printed platform designed to precisely position iontronic devices in standard 96-well transwell plates. **c** Cell viability of U-87 cells after operating iontronic devices delivering **1** for 4 h at +20 nA (Active, *n* = 8) and at −20 nA (Reverse, *n* = 7)

into transwells holding CA4-immobilized beads from (**a**). Points represent individual biological replicates; bars indicate mean ± SD. GC growth control (untreated, *n* = 32), "No Beads" = operation of iontronic devices delivering **1** at +20 nA, no beads in transwell present (*n* = 6), "No Tz **1**" = CA4-immobilized beads in transwell present with iontronic delivery of K+ (0.1 M KCl) instead of Tz **1** (*n* = 8). Statistical analysis was done *via* a Brown–Forsythe and Welch ANOVA tests (one-way) and Dunnett's T3 multiple comparisons test. $\alpha$ = 0.05. *p*-values: GC vs. No Beads (ns) = 0.2377; GC vs. No Tz **1** (ns) = 0.3827; GC vs. Reverse, 4 h (ns) = 0.5739, GC vs. Active, 4 h (****) = <0.0001. Certain graphical elements in (**a**) were created in BioRender. Hecko, S. (2026) https://www.BioRender.com/z8yr1eu.

## Iontronic release of immobilized CA4

Building on these results, we employed cTCO-bis-NHS (**5**) to synthesize biotin-cTCO-DMEDA-CA4 (**6**), enabling immobilization of the prodrug onto streptavidin-coated magnetic beads, which were selected as a well-defined solid support allowing quantitative analysis of release while serving as a simplified proof-of-principle model system. We used the discrepancy in reactivity of the NHS-carbonate and NHS-ester groups (Fig. 4a), which allowed for the sequential decoration of the cTCO-linker[52]. This strategy facilitated the rapid synthesis of **6** using DMEDA, biotin-NH2, and subsequent reaction with CA4-*p*-nitrophenol (CA4-*p*NP) (Supplementary Information 2). The biotin moiety allowed **6** to attach onto magnetic streptavidin-coated beads (Dynabeads MyOne™ Streptavidin C1). Beads were co-incubated with U-87 cells in a transwell setup, and iontronic devices for delivery of **1** were positioned by immersing the device tip just above the beads within the transwell inserts (Fig. 4a, right). To facilitate consistent positioning of the iontronic devices within the 96-well transwell plates, a custom 3D-printed platform was developed (Fig. 4b). This setup accommodated the complex geometry of standard high throughput screening (HTS) transwell supports by incorporating a funnel-like mounting structure, offering precise placement of the device's outlet in the top well and the counter electrode in the bottom compartment where the cells were cultured. The funnel serves as an agar bridge electrode and was filled with agarose hydrogel to maintain ionic contact while electrically isolating the electrode from direct contact with the cell culture environment, thereby reducing cytotoxic effects stemming from Ag/AgCl electrodes.

The amount of bead-immobilized prodrug **6** corresponded to a nominal concentration of 1 μM. Devices were operated for periods of

4 h in forward and reverse at ±20 nA (potential traces shown in Supplementary Figs. S14–16) followed by removal of the pumps. All wells were then incubated under static conditions for 72 h, followed by cell viability determination (Fig. 4c). Active delivery of **1** at +20 nA released **CA4** from beads, which resulted in a significant reduction of cell viability to 24%, corresponding to the maximum toxic effect of **CA4**. In comparison, the reverse operation at −20 nA had no significant effect on cell viability, nor did operating devices at +20 nA without beads or the presence of beads with bound prodrugs without device operation.

## Iontronic release of immobilized BSA

To verify that iontronic click-to-release can be applied to payload delivery independently of molecular weight, we demonstrated release of immobilized **BSA** (66 kDa) using the identical bead-based model system (Fig. 5a). Therefore, a bicyclo[6.1.0]non-4-yne (BCN) moiety was installed analogously to the synthesis of **6**, which enabled the covalent attachment of commercial mono-functionalized BSA-azide (**BSA-N3**) to afford biotin-cTCO-BSA (**7**, see Methods) *via* strain-promoted azide-alkyne cycloaddition (SPAAC). The mono-functionalization of BSA ensured site-specific conjugation and a defined 1:1 stoichiometry between protein and cTCO. The beads were subsequently coated with **7**. Iontronic devices were immersed in PBS-buffered bead suspensions and operated for 1–4 h (potential traces shown in Supplementary Figs. S17–19), followed by removal of the pumps and incubation under static conditions for 24 h. Protein release was subsequently analyzed *via* SDS-PAGE, with bands visualized using InstantBlue® and quantified using ImageJ (v1.54p) software (Supplementary Information 8.2). The amount of protein released increased

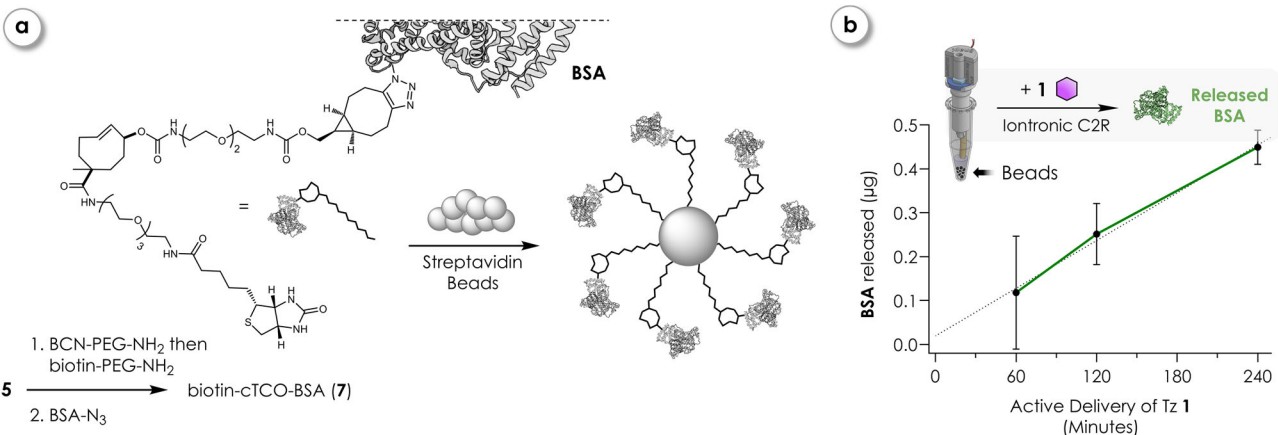

**Fig. 5 | Iontronic click-to-release of bead-immobilized BSA. a** Schematic of the synthesis and immobilization strategy for biotinylated click-cleavable bovine serum albumin **7** immobilized on streptavidin-coated magnetic beads. **b** BSA release as a function of iontronic pumping duration of Tz **1**, quantified by SDS-PAGE and ImageJ analysis. Data are shown as mean ± SD of $n = 4$ independent experiments per time point. Iontronically induced click-to-release (C2R) leads to time-dependent release of **BSA**; linear regression analysis indicates a release rate of ~0.11 μg h⁻¹ at +20 nA active delivery. As in Fig. 4, grey **BSA** indicates the "inactive" bound state, while green (colored) indicates an "active" released state. Certain graphical elements in this figure were created in BioRender. Hecko, S. (2026) https://www.BioRender.com/z8yr1eu.

linearly with pumping duration, with a release rate of 0.11 μg (~1.5 pmol) **BSA** per hour of active delivery of **1** at +20 nA (Fig. 5b).

When delivering K⁺ (0.1 M KCl) as a control, we observed a background level of **BSA** release, which we attributed to passive protein dissociation or electrically induced detachment from the bead surface. This background was subtracted to isolate the iontronic click-to-release component, and the corresponding data are provided in Supplementary Information 8.3. These assay-related effects arise from the bead-based platform and are anticipated to be attenuated in future implementations using hydrogel-based depot architectures.

## Discussion

This work describes a class of electrochemical interfaces that function as chemical switches translating electrical input into targeted molecular release. In contrast to conventional iontronic- or iontophoretic drug delivery, where drug molecules are transported in their native form, this approach uses electrophoretic pumping of aminoethyl Tz **1**, which triggers the cleavage of TCO-bound, immobilized payloads. This strategy decouples the molecular constraints of the transport from the physicochemical properties of the drug, while retaining the precise spatiotemporal control over release. The delivery rate of **1**, and consequently the payload release, scaled linearly with applied current, offering programmability and reversible ON/OFF control. We validated this concept across multiple payloads and formats. First, we achieved controlled release of the highly potent chemotherapeutic agent combretastatin A-4 (**CA4**) from its prodrug form *via* Tz **1** delivery, resulting in targeted cytotoxicity against human glioblastoma U-87 cells. We then immobilized the prodrug on functionalized beads and showed equally potent, localized drug release through iontronic C2R. Finally, to demonstrate compatibility with large biomolecules well beyond the iontronic transport (size) limit, we triggered the release of bovine serum albumin (**BSA**, 66 kDa) from bead scaffolds, with released amounts scaling linearly with Tz pumping duration. This confirmed the platform's ability to electronically control the release of both small- and macromolecular therapeutics independent of their charge or size. Low-voltage operation, combined with the small form factor of IPs, opens the possibility for integration into embedded systems and facilitates translation toward minimally invasive, clinically relevant in vivo applications. While beads were used here for proof-of-concept due to their ease of functionalization and commercial availability, future implementations will employ biocompatible and injectable scaffolds such as hydrogels or soft polymer matrices. The presented approach provides a framework for electroceuticals, where electronically orchestrated drug release profiles can be tailored to individual patients, disease states, and circadian rhythms.

## Methods

### Device fabrication

Sections of polyimide-coated fused silica capillaries (CM Scientific, TSP025150-CFPH01AA, 25 μm inner diameter, 150 μm outer diameter), with a length of 500 mm, were attached to EFD precision tips (Nordson, 20FA TT 0.23") by melting the plastic tips while the capillaries were inserted approximately 0.5 cm. The EFD precision tips were attached to syringes that were vertically connected to a N₂ supply. The inner glass surface of the capillaries was etched by flushing with 2 M KOH for 2 h using N₂ at a pressure of 2 bar. Next, the capillaries were washed with deionized water (10 min) and dried with N₂ (10 min). An adhesion promoter (3-(trimethoxysilyl)propyl methacrylate, A 174, 10 wt% in toluene, Sigma Aldrich) was flushed for 5 h. The capillaries were flushed with EtOH (20 min) and dried with N₂ (10 min). Finally, the capillaries were filled with the polyelectrolyte mixture. The polyelectrolyte mixture consisted of 2-acrylamido-2-methylpropane sulfonic acid sodium salt (AMPSA, Sigma Aldrich), poly(ethylene glycol) di-acrylate Mn 575 (PEGDA, Sigma Aldrich), and photoinitiator lithium phenyl-2,4,6-trimethylbenzoylphosphinate (LAP, Sigma Aldrich). The components were mixed in deionized water to achieve an 85:15 mol% AMPSA:-PEGDA ratio with 0.1 mol% LAP photoinitiator at a total concentration of 25 wt%. Prior to photo-exposure, the capillaries filled with polyelectrolyte were cut from the plastic tips into pieces of approximately 150 mm length. They were mounted onto a plate using double-sided tape and the ends of the capillaries were sealed with tape. The capillaries were placed in a custom-made photo-exposure box (UV-A, 3 mW cm⁻²) and the CEM was polymerized for 22 h. The capillaries were further cleaved into pieces of ~15 mm with a fiber cleaver system (Fujikura CT-02) and stored in 0.1 M KCl.

### Iontronic device operation

**Constant-current mode.** Iontronic devices were assembled by inserting the filled 15 mm capillaries into custom 3D-printed reservoirs and mounting them onto Eppendorf tubes (0.5 mL) or 96-well plates containing a target reservoir of 40–100 μL 1× PBS (Sigma Aldrich). The source reservoir was filled with 5 mM **1** in water. To operate the IPs, electrodes (Ag/AgCl) were placed in both target and source

electrolytes. The IPs were operated at constant current using an OctoStat30 (Ivium Technologies). To load the IPs with **1**, the IPs were operated at +20 nA for 16–24 h. Complete loading of the IPs was indicated by a plateau in voltage (Supplementary Fig. S5).

**Step-function mode.** The IPs were assembled in the same manner as previously described, utilizing a 3D-printed holder that was adapted to accommodate a larger volume of target reservoir of 150 μL. The input signal was alternated periodically through levels of +20 nA ("ON") for 16 h and zero current ("OFF") for 7 h. A short reverse bias step (1 h, −20 nA) was incorporated between the ON and OFF steps to facilitate the retention of **1** within the channel and prevent passive diffusion in the OFF state.

### Iontronic delivery of tetrazine 1

Delivery rates of **1** (in pmol min$^{-1}$) were determined by operating iontronic devices at constant current (e.g., 0, +10, +20, −20 nA for set periods, typically 2–16 h). The concentration of delivered **1** in the target solution was measured using a fluorogenic assay *via* the C2R of 4-methylumbelliferone (**4MU**) from rTCO-DMEDA-4MU (**2**). Fluorescence measurements were performed on a plate reader (Synergy H1 microplate reader, BioTek) at 25 °C. The rTCO-DMEDA-4MU (**2**) probe was prepared as a 500 μM stock solution in DMSO. Tetrazine samples collected post-IP operation were diluted 1:1 into 20 × PBS (8 μL sample + 8 μL 20 × PBS) to reach a final buffer concentration of 10 × PBS. The C2R reaction was initiated by adding 4 μL of the rTCO-DMEDA-4MU (**2**) stock solution to the diluted Tz samples in a 384-well microplate. The fluorescence intensity ($\lambda_{ex} = 360$ nm, $\lambda_{em} = 480$ nm) was recorded over a period of 4 h at 25 °C. Tetrazine concentrations were determined from external calibration curves generated by reacting **2** (100 μM) with a series of known Tz **1** concentrations (5, 10, 50, 100 μM). The fluorescence intensities after 4 h were used to determine tetrazine concentrations. The amount of delivered Tz **1** and charge as depicted in Fig. 2c was calculated as follows:

$$Delivered\ tetrazine\ \mathbf{1}\ (nmol) = conc.\ \mathbf{1}\left(mol \cdot L^{-1}\right) \cdot target\ volume\ (L) \cdot 10^9$$

(1)

$$Charge\ (mC) = current\ (nA) \cdot delivery\ time\ (s) \cdot 10^{-6}$$

(2)

Delivery rates in Fig. 2e were calculated using the following equation:

$$Delivery\ rate\ of\ \mathbf{1}\left(pmol\ min^{-1}\right) = \frac{conc.\ \mathbf{1}\left(mol \cdot L^{-1}\right) \cdot target\ volume\ (L) \cdot 60\left(\frac{s}{min}\right)}{delivery\ time\ (s)} \cdot 10^{12}$$

(3)

The electrochemical efficiency was calculated by relating the experimentally delivered amount of Tz **1** to the theoretical amount of **1** defined by the total charge delivered using the following equation:

$$Electrochemical\ efficiency = \frac{delivered\ Tz\ \mathbf{1}\ (mol)}{\left(\frac{delivered\ charge\ (C)}{2 \cdot F}\right)} \cdot 100\%$$

(4)

Here, F denotes Faraday's constant (96 485 C mol$^{-1}$), and a charge number of 2 is used to account for the double charge of Tz **1**.

### Preparation of CA4 beads

Dynabeads MyOne™ Streptavidin C1 (10 mg/mL, Thermo Fisher Scientific) were used for conjugation. A 100 μL aliquot of the bead suspension was washed three times with phosphate-buffered saline (PBS) using a DynaMag™-2 magnetic rack (Thermo Fisher Scientific) for separation. The washed beads were resuspended in 1 mL of PBS containing 5% (v/v) DMSO. To this suspension, 1 μL of biotin-cTCO-

DMEDA-CA4 (**6**) (9 mM in DMSO) was added. The concentration of the cTCO stock solution was determined by absorbance titration using 3,6-bis(2-pyridyl)tetrazine (**2Pyr₂**, Sigma Aldrich), as described in Supplementary Information 3.1. Conjugation was performed at room temperature for 1 h under continuous gentle agitation. Following incubation, the **CA4**-functionalized beads were sequentially washed: twice with 1 mL of 5% DMSO in PBS, three times with PBS, and twice with Eagle's minimum essential medium (E-MEM, Sigma Aldrich) containing 10% fetal bovine serum (FBS), 4.5 g/L glucose, 2 mM L-glutamine, and 1% MEM Non-Essential Amino Acids. Beads were finally resuspended in the same supplemented E-MEM to a final concentration of 1 mg/mL for use in iontronic **CA4** release experiments.

### Preparation of BSA beads

Mono-functionalized azide-modified bovine serum albumin (**BSA**-N₃, 0.1 mg/mL in PBS, 200 μL, Vector Laboratories) was reacted with biotin-cTCO-BCN (**S3**) (10 μL, 87 mM in DMSO; 300-fold molar excess) under static conditions for 1 h at room temperature. Prior to the addition of the TCO, the concentration of the cTCO stock solution was determined by absorbance titration with **2Pyr₂**, following Supplementary Information 3.1. The reaction mixture was purified using Zeba™ Spin Desalting Columns (0.5 mL, Thermo Fisher Scientific) to remove unreacted **S3**. If residual **S3** was detected by HPLC-MS, the purification step was repeated. For bead conjugation, 1 mL of Dynabeads MyOne™ Streptavidin C1 (10 mg/mL, Thermo Fisher Scientific) was used per 100 μL of biotin-cTCO-BSA (**7**) solution. Owing to the reduced binding capacity of biotinylated proteins on Dynabeads MyOne™ Streptavidin C1 beads, the quantity of beads was increased tenfold relative to those used in the **CA4**-release experiments. Beads were washed three times with PBS using magnetic separation (DynaMag™-2), then resuspended in 800 μL PBS. Subsequently, 200 μL of the BSA-conjugate **7** solution was added. The mixture was incubated for 1 h at room temperature under continuous gentle agitation. Following incubation, the BSA-bead complexes were washed five times with 1 mL PBS and resuspended in PBS to a final bead concentration of 10 mg/mL for use in iontronic **BSA** release experiments.

### Cell culture and assays

The human GBM cell line U-87 (provided by the MUG Cell Bank, catalogue no. 300367, CLS) was cultured at 37 °C and 5% CO₂ in Eagle's minimum essential medium (E-MEM) containing 10% fetal bovine serum (FBS), 4.5 g/L glucose, 2 mM L-glutamine, and 1% MEM Non-Essential Amino Acids. All assays were performed using flat-bottomed, transparent 96-well plates (Corning). For iontronic release studies using **CA4**-beads, transparent HTS transwell 96-well plates with a pore size of 0.4 μm (Corning) were used. For cell proliferation studies, U-87 cells (6000 cells/well) were treated for 72 h with freshly prepared compound solutions: sulfo-cTCO-DMEDA-CA4 (**4**, 0.01–1000 nM), **CA4** (0.01–1000 nM), **4** (0.01–1000 nM) + **1** (5 μM), **1** (5 μM). For iontronic release studies, penicillin/streptomycin was added to the media to a final concentration of 1% v/v. Cell viability was determined by MTS assay (CellTiter 96® Aqueous One Solution Cell Proliferation Assay Promega, Germany) according to the manufacturer's protocol and was performed on a CLARIOstar plate reader (BMG Labtech, Germany) at 490 nm with top optics and orbital averaging. IC₅₀ values were obtained by fitting data points to a sigmoidal curve using Prism 10 (GraphPad software, USA) and the following equation, in which top and bottom refer to the minimally and maximally inhibited response, respectively:

$$Y = Bottom + \frac{Top - Bottom}{1 + \left(\frac{IC_{50}}{X}\right)^{HillSlope}}$$

(5)

## Iontronic release of CA4 from sulfo-cTCO-DMEDA-CA4 (4) in cellular environment

Tetrazine **1** (5 mM) was loaded into the source reservoir of the iontronic devices (AMPSA/PEGDA 85/15 mol% ratio at 25 wt% in de-ionized water, 15 mm length, 25 μm inner diameter, 150 μm outer diameter, Ag/AgCl-source electrode). Devices were operated at constant current using an 8-channel OctoStat30 (Ivium Technologies). The channel loading was done at +20 nA over 15 h. Agar bridge electrodes were prepared by adding agar (4 wt% in 0.1 mM KCl) into custom 3D-printed funnels and covering the agar with aqueous KCl (0.1 mM). Above the agar, a freshly etched Ag/AgCl electrode was placed. Agar bridge electrodes were sterilized by 30 min UV irradiation in a laminar flow hood. 6000 U-87 cells were seeded in 96-well plates and incubated in 250 μL cell media containing 100 nM sulfo-cTCO-DMEDA-CA4 (**4**). Iontronic devices were installed in the cell wells, together with an agar bridge Ag/AgCl. Devices were operated at +20 nA for active delivery, or at −20 nA for reverse delivery for 1 h each. Afterwards, devices and agar bridges were removed, and plates were incubated at 37 °C and 5% $CO_2$ and cell viability was read out using a MTS assay after 72 h.

## Iontronic release of CA4 from beads

Iontronic devices (AMPSA/PEGDA 85/15 mol% ratio at 25 wt% in de-ionized water, 15 mm length, 25 μm inner diameter, 150 μm outer diameter, Pt-source electrode) were set up the same way as for the iontronic release of **CA4** from sulfo-cTCO-DMEDA-CA4 (**4**) on cells (source reservoir concentration: 5 mM **1**, agar bridge electrodes, 4 wt% in 0.1 mM KCl; Ag/AgCl). U-87 cells (6000 cells/well) were seeded in the transwell 96-well receiver plates and incubated in 150 μL cell media. Transwell insert plates containing 50 μL of **CA4**-bead suspension per well were submerged and devices installed subsequently. The remaining wells were closed with custom-made lids to prevent contamination and evaporation. Devices were operated at +20 nA for active delivery or at −20 nA for reverse delivery for 4 h each and were removed after operation. Subsequently, plates were incubated at 37 °C and 5% $CO_2$ for 72 h and cell viability was determined as described above.

## Iontronic release of BSA from beads

Iontronic devices (AMPSA/PEGDA 85/15 mol% ratio at 25 wt% in deionized water, 15 mm length, 25 μm inner diameter, 150 μm outer diameter; source reservoir concentration: 5 mM **1**) were installed on 0.5 mL Eppendorf tubes using the custom 3D-printed housings. Pt-electrodes were integrated into the device design (Fig. 2b). The channel loading was done at +20 nA over 13.6 h. After reaching a steady potential indicating a fully occupied CEM, iontronic devices were installed on 0.5 mL Eppendorf tubes containing 50 μL BSA-bead suspension and run at +20 nA for different time periods (1, 2, and 4 h). Delivery of $K^+$ (0.1 M KCl in the source reservoir) was performed for 4 h instead of **1** to isolate the contribution from passive or electrochemically induced detachment of **BSA** from the beads. Similarly, beads were incubated for 4 h without any iontronic intervention. After 24 h following active delivery, the magnetic beads were separated from the supernatant using the DynaMag™ -2 magnetic rack and all **BSA**-containing samples were stored at −20 °C. For SDS-PAGE analysis, 10 μL of **BSA**-containing samples were mixed with 3.33 μL of Bio-Rad 4× Laemmli sample buffer (Bio-Rad, #1610747) and heated to 90 °C for 10 min. Samples were then loaded onto precast Novex™ WedgeWell™ 12% Tris-Glycine gels (Thermo Fisher Scientific, XP00120BOX) and electrophoresed at 200 V for 45 min using a Bio-Rad PowerPac™ HC system. Gels were stained with InstantBlue™ Coomassie protein stain (Abcam, ab119211) for 1 h without destaining. Imaging was performed using a Bio-Rad ChemiDoc™ imaging system. Band intensities were quantified using ImageJ software (v1.54p), and protein concentrations were determined by comparison to a **BSA** calibration curve (0.05–1.0 μg) run on the same gels.

## Statistical analysis

Statistical analysis was performed using commercially available software (Prism 10, GraphPad, USA) and specific statistical tests were used as stated in the figure description.

## Ethics

This study does not involve experiments involving animals, human participants, or clinical samples.

## Reporting summary

Further information on research design is available in the Nature Portfolio Reporting Summary linked to this article.

## Data availability

All data supporting the findings of this study are available within the article and its supplementary files. Processed and raw data files generated in this study, including quantitative datasets underlying the figures and raw NMR spectroscopy data, have been deposited in the public institutional repository of TU Wien (TU Wien Research Data) under https://doi.org/10.48436/vxpen-9kg70 (https://doi.org/10.48436/vxpen-9kg70). Any additional requests for information can be directed to and will be fulfilled by the corresponding authors. Source data are provided with this paper.

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

## Acknowledgements

This work has received funding from the European Union's Horizon Europe Research and Innovation Programme under Grant Agreement 101099963 (bioSWITCH—M.B., D.T.S., R.S., L.W., H.M., J.B.) and the European Research Council under Grant Agreements 101117736 (Time2SWITCH—J.B.) and 101042881 (bioTARGET—H.M.). This research was funded in part by the Austrian Science Fund (FWF) [10.55776/J4304—J.B., 10.55776/I4623—H.M., 10.55776/Y1443—H.M.]. Additional funding was provided by the Knut and Alice Wallenberg Foundation (M.B., D.T.S., T.A.S.). We gratefully acknowledge the support of the PhD program Molecular Medicine (MOLMED) at Medical University of Graz (C.B.). L.W. received funding from the BioTechMed Graz Young Research Groups. We wish to thank Aneta Marková, Loïc Baudoin, and Peter Nadj for their assistance in the early development of multi-well iontronics, and Maria Seitanidou, Tobias Abrahamsson and David J. Poxson for assistance with preliminary tetrazine delivery experiments, and Adam El-Zoghbi for help with tetrazine synthesis. The authors acknowledge TU Wien Bibliothek for financial support through its Open Access Funding Programme.

## Author contributions

S.H. synthesized, characterized, and applied compounds for immobilization and the subsequent iontronic click-to-release (C2R) of CA4 and BSA, optimized prodrug activation and protein release assays, and drafted the manuscript, including figure preparation. M.E.J.V. optimized ion exchange membrane composition, developed and optimized protocols for ion exchange membrane fabrication, assembled iontronic devices, performed and optimized tetrazine delivery and step-function experiments, conducted assay quantification, data analysis, figure preparation, and contributed to manuscript writing. C.B. performed iontronic click-to-release (C2R) experiments with sulfo-cTCO-DMEDA-CA4 in solution and on beads, assisted with BSA release experiments, and contributed to data analysis, figure preparation, and manuscript writing. D.B. designed, fabricated, and optimized custom 3D-printed components, implemented device optimization strategies, performed tetrazine delivery experiments, supported data analysis and figure preparation, and contributed to manuscript writing. M.E.H. improved and implemented IEM fabrication protocols, assembled iontronic devices, developed and improved characterization setups, conducted tetrazine delivery and step-function experiments, and contributed to data analysis, figure preparation, and manuscript writing. N.P. synthesized and characterized fluorogenic rTCO probes and tetrazines, carried out iontronic tetrazine delivery experiments, assay development, and cyclic voltammetry measurements. R.B. supported 3D-printed component fabrication, device optimization, and tetrazine delivery, performed step-function tuning, and contributed to data analysis, figure preparation, and manuscript writing. P.K. synthesized and characterized cTCO-prodrugs and conducted all HPLC-MS-based release studies and contributed to manuscript writing. A.L. performed kinetic measurements for the C2R reaction. W.K. synthesized the multi-step cTCO linker. H.S.U. designed ion exchange membrane composition, developed iontronic device fabrication methods, designed and fabricated initial custom 3D-printed components. I.B.W. designed ion exchange membrane composition, developed and supervised iontronic device fabrication, designed initial characterization setup. T.A.S. designed ion exchange membrane composition, designed initial characterization setups, provided supervision, and secured funding. M.B. provided project supervision, secured funding, and contributed to manuscript writing. D.T.S. provided project supervision, secured funding, and contributed to manuscript writing. R.S. provided project supervision and contributed to manuscript writing. L.W. performed preliminary iontronic in vitro C2R experiments with sulfo-cTCO-DMEDA-CA4, designed in vitro studies, provided project supervision, and contributed to manuscript writing. H.M. conceived the study, co-led project supervision, developed the bioorthogonal chemistry and synthetic strategy, secured funding, and co-wrote the manuscript. J.B. conceived the study, co-led project supervision, developed the iontronic-C2R delivery concept, performed early C2R iontronic experiments, secured funding, and co-wrote the manuscript.

## Competing interests

I.B.W., T.A.S., M.B., D.T.S., H.M., and J.B. are shareholders in OBOE IPR AB, which owns patents related to this research and is the parent company of Iontronics AB. OBOE IPR AB is the applicant of the European patent EP-4114508 B1 (granted), which covers iontronic delivery systems relevant to the work reported in this manuscript. J.B., H.M., D.T.S., and M.B. are named inventors on this patent. The remaining authors declare no competing interests.
