## [Peer Review file · Nature Communications]

Iontronic Click-to-Release Enables Electrically Controlled Delivery of Drugs and Biomolecules Beyond Charge and Size Limitations

Corresponding Author: Professor Johannes Bintinger

Version 0:

Reviewer comments:

Reviewer #1

(Remarks to the Author)

Abstract

I believe the abstract gives a clear and honest overview of the study. In my view, it already covers everything needed. The only line that feels slightly overstated is the claim about “next-level electrochemical devices” — I would suggest toning that down a bit.

Introduction

Crucially, the introduction then introduces the novel solution in a logical way. The authors explain that by integrating iontronic delivery with bioorthogonal click-to-release (C2R) chemistry, they decouple the transport mechanism from the payload activation. In my view this is a clear statement of novelty. The text and the schematic in Fig. 1 together make it easy to understand that instead of pumping the drug itself, they pump a chemical trigger (a charged tetrazine) which then reacts with an immobilized prodrug to release the actual drug.

If I were to suggest any additions, it would be to perhaps cite or acknowledge any closely related prior attempts (if any) at remotely triggered drug release using chemical reactions. The authors have cited bioorthogonal prodrug release studies, but it might further strengthen novelty to explicitly contrast their approach with other stimulus-responsive drug delivery systems. For example, mentioning how iontronic click-to-release differs from light-triggered or magnetically triggered release systems (in terms of precision or invasiveness) could be beneficial. But, this is a minor point.

Programmable tetrazine delivery

I believe the correlation between charge and delivered tetrazine is pretty convincing to me, but the electrochemical efficiency is not fully addressed (I may have missed that part). It would strengthen this section if the authors reported at least an approximate charge-to-molecule conversion ratio. Without that information, it is difficult to assess how much of the applied current contributes to actual ion transport versus leakage. But I understand these may not have been part of the study's design. Even so, the strong linearity between applied charge and delivered amount already supports the claim of precise electrical control. Adding a short statement acknowledging this limitation would make the section feel complete and transparent.

The linear charge–delivery trend is convincing, yet the error spread in Figure 2c appears uneven and is not discussed. Authors may consider specifying n and providing R^2 to clarify how consistent the linear response is across devices.

CA4 Prodrug Activation

This section gives us good chemical evidence through HPLC that the CA4 prodrug can be efficiently cleaved in solution. However, it seems to me this analysis reflects bulk reaction conditions (please correct me if I am mistaken), while the later cell-based results, for example Figure 3c, only infer successful release indirectly through cytotoxicity. As a personal question, don't authors think that, without chemical verification, it remains not more than an assumption that the same release mechanism occurs in the iontronic setup? Including or referencing a chemical confirmation of CA4 formation under iontronic operation would bridge these two pieces of evidence and make the mechanistic link between electrical input and molecular release more realistic?

CA4 Release from Beads (Figure 4)

Here authors performed an experiment that uses a fixed 4-hour stimulation period at ± 20 nA. This may show On/off functionality but the degree of dose programmability kind of feels uncertain. However, it would help if the authors explained whether shorter activation times or lower currents were tested and yielded some certain effects.

In Figure 5b, we can see a background release of BSA and this is shown in red in the figure. Authors wrote:

“When KCl solution was pumped instead of tetrazine, a substantial release of BSA was still detected, which we attribute to an electrochemically induced bead detachment effect.”

Could you please expand this discussion a bit. I think this part is critical.

Other than the specific points discussed above, the figures are clear, well organized, and visually engaging; the authors managed to make a complex process easy to follow. The work demonstrates high novelty by integrating ionotronic delivery with click-to-release chemistry. Addressing the controlled release of both small and macromolecules was important. The dataset is rich. The manuscript can be accepted after addressing aforementioned minor points.

Reviewer #2

(Remarks to the Author)

The manuscript co-lead by Waldherr, Mikula, and Binting demonstrates proof-of-concept for ionotronic click-to-release chemistry of prodrugs and proteins. I enjoyed reading this manuscript, which spans the fields of electroceuticals, bioorthogonal and prodrug chemistry. The manuscript clearly describes precise spatiotemporal control using a model prodrug and masked protein (BSA), thus, will be of great interest to the chemical biology and medicinal chemistry fields. It is also pitched to a level that is accessible for other scientists, especially those in the electrochemical and materials space.

The methods are described in detail with high quality experimental data that supports the conclusions of the manuscript, i.e., to demonstrate spatiotemporal control of a bioorthogonal click-to-release reaction, and this manuscript will provoke interest in the area.

The proof-of-concept was demonstrated via prodrug (CA-4) activation or protein unmasking using well-established tetrazine-TCO click-to-release chemistry, which itself is not novel, but is novel and exciting when combined with electrochemically driven spatiotemporal prodrug activation. The setup enables a controlled and pulsatile release of chemical activator (tetrazine) from a cleverly designed reservoir which then reacts with a TCO-prodrug (or protein) that leads to the tautomeric diazine and release steps. The tetrazine is held in the reservoir that has an ion exchange membrane trapping the cationic tetrazine activator molecule on one side, and on the other side, payload (prodrug or enzyme) immobilised on streptavidin beads (via a biotin attached to the TCO-payload). The current is applied, and the membrane is then permeable to the tetrazine. Reversal of the current was used to demonstrate no permeability of the tetrazine, proving that the ionotronic pump and an applied current is essential to achieve payload activation.

Cell viability of the prodrug alone, and immobilised on beads, showed comparable results, paving the way for similar systems that are more bio-relevant than the beads used in the proof-of-concept (e.g., hydrogels). Ultimately, this approach will evolve into a system that could be used in vivo (mentioned in the manuscript conclusions) and is the first step to therapeutic implants that can adapt to changes in real-time, being switched off and back on again through application of an electrical current.

Something that is not considered here is the lack of regeneration of the depot's payload. Essentially, the TCO is consumed with each reaction and eventually would be exhausted from the depot, rendering it ineffective to further tetrazine released from the ionotronic pump. However, this is another challenge, and it will be exciting to see the next generation of this system and its application to in vivo implants and prodrug activation strategies.

The following points should be considered by the authors during any revision:

The NMR spectra and reported data are good and of very high purity, but there are some NMR assignments that should be corrected (details below).

-Compound 2 has substantially more ^{13}C peaks assigned than is expected. Are these due to rotamers? Also, in the spectra for Compound 2, there are 7 carbon peaks in the 130-185 ppm region (page 28, ESI) but from the structure, I would expect 6 carbon peaks (4 x carbonyl and 2 x alkene).

-Compound S3 has 44 assigned carbon peaks, but only 42 expected. Looking at the spectra it appears there are 42 peaks (page 30), so the authors must have assigned extra peaks.

-Compound 2 and S3 have carbon assignments that are identical (e.g., 70.6, 70.6, 70.3 x 3, etc.). These peaks that are identical should be assigned to 2 decimal places, to distinguish them from each other.

-It is difficult to see using the spectra provided, but some of the coupling constants (J values) for compounds are very large and may not be assigned correctly. Compound 2 is an example where J values of 33 Hz and 22 Hz are reported. Are these correct? Should these peaks that are very complex be assigned as multiplets (m)?

Reviewer #3

(Remarks to the Author)

Claim: The authors demonstrated an iontronic system that deliver charged tetrazines as molecular scissors to selectively click trans-cyclooctane linkers and release tethered molecules on demand. They showed that electronically tunable tetrazine delivery sustained over several days and precise release of payloads ranging from small molecules to large proteins. By integrating biorthogonal click-to-release chemistry with iontronic, the current work overcomes molecular size, charge and stability limitations.

Strengths of the manuscript:

- (i) Click-to-release is an interesting concept that beautifully utilizes charged tetrazines as molecular scissors to cleave the bond.
- (ii) This concept overcomes the limitation of delivering large molecular weight payload.
- (iii) The molecular conjugates were thoughtfully designed and characterized. The experimental demonstrations are clear in most cases.

Weaknesses of the manuscript

- (i) The concept is solely dependent on molecular design. The author has shown only one transporter molecule i.e. tetrazine.
- (ii) Solubility of the conjugate molecule might be a major concern.

Comments: Accepted after major revision required.

1. The release kinetics of the payload are likely governed by the formulation of the cation-exchange membrane (PEGDA/AMPSA), since crosslink density and membrane thickness strongly influence diffusion and how long tetrazine is retained in the matrix. Did the authors evaluate these effects (PEGDA:AMPSA ratio, degree of polymerization/crosslinking, and film thickness)?

A plot showing component concentration (or PEGDA:AMPSA ratio / crosslink density) versus tetrazine retention time (with membrane thickness as a separate series or secondary axis) may clarify how membrane composition can control release kinetics.

2. The rationale for using beads is not clearly explained in the manuscript. Could the authors clarify the choice. Does the size of the beads influence the outcome? I guess it will.

3. In figure 2e, for a constant current of +5 nA, the delivery rate was 0.7 ± 0.6 pmol min⁻¹. This value exhibits a very large relative error. Comment on whether the applied current was too insignificant to trigger any mobility.

4. The second-order rate constant ($46 \text{ M}^{-1}\text{s}^{-1}$) was determined using a model cTCO compound. How well do you expect this kinetic value to translate to the actual prodrug (compound 4) in the cellular environment, where diffusion and steric factors differ?

5. Did you examine the stability of the prodrug in cell culture medium or serum over time (without Tz), to confirm no slow, non-specific release of CA4 occurs?

6. The covalent attachment of biotin-cTCO to BSA via SPAAC is clear. However, it would be valuable to report the average number of cTCO groups per BSA molecule (degree of functionalization) and additionally any potential protein denaturation during SPAAC attachment or immobilization could affect release and activity should be addressed.

Version 1:

Reviewer comments:

Reviewer #1

(Remarks to the Author)

In my view, the authors have adequately addressed all of the concerns I raised, and I believe the manuscript is suitable for publication without the need for further revisions.

Reviewer #2

(Remarks to the Author)

As in my initial review, this is an excellent manuscript reporting on proof-of-concept for iontronic click-to-release chemistry of prodrugs and proteins. The authors have provided a detailed response to all reviewer comments. In my opinion, they have addressed the comments well and have made the necessary adjustments to the manuscript, including the characterization data requested by this reviewer.

Reviewer #3

(Remarks to the Author)

I am happy with the revised manuscript as the authors have successfully addressed the concerns.

Point-by-Point Response to Reviewer Comments

Iontronic Click-to-Release Enables Electrically Controlled Delivery of Drugs and Biomolecules Beyond Charge and Size Limitations

submitted to Nature Communications

We thank the reviewers for their careful evaluation and constructive feedback. We have revised the manuscript to address all comments. Below, we provide a detailed, point-by-point response indicating the changes made to the text. Reviewer comments are shown in *italics*, followed by our responses.

Reviewer #1

General Comment:

... the figures are clear, well organized, and visually engaging; the authors managed to make a complex process easy to follow. The work demonstrates high novelty by integrating iontronic delivery with click-to-release chemistry. Addressing the controlled release of both small and macromolecules was important. The dataset is rich. The manuscript can be accepted after addressing aforementioned minor points.

Response

We appreciate Reviewer #1's detailed and positive assessment of our work and their helpful suggestions for improving clarity and completeness. We have carefully addressed each of the specific points raised and revised the manuscript accordingly.

Comment #1

I believe the abstract gives a clear and honest overview of the study. In my view, it already covers everything needed. The only line that feels slightly overstated is the claim about "next-level electroceutical devices" — I would suggest toning that down a bit.

Response

We have revised the phrase in the *Abstract* to "**advanced** programmable electroceutical devices".

Comment #2

If I were to suggest any additions, it would be to perhaps cite or acknowledge any closely related prior attempts (if any) at remotely triggered drug release using chemical reactions. The authors have cited bioorthogonal prodrug release studies, but it might further strengthen novelty to explicitly contrast their approach with other stimulus-responsive drug delivery systems. For example, mentioning how iontronic click-to-release differs from light-triggered or magnetically triggered release systems (in terms of precision or invasiveness) could be beneficial. But, this is a minor point.

Response:

We performed two edits in the *Introduction*. First, we adapted and expanded the paragraph on implantable systems to include magnetic, thermal, and ultrasound-activated devices, emphasizing their limited spatial precision, potential invasiveness, and lack of repeatable dose control:

Implantable devices and osmotic pumps have been successfully designed to deliver drugs locally^{4,13-20}, but these technologies typically rely on pre-programmed release or external triggers. Such triggers can be chemical (e.g., pH changes/hydrolysis^{21,22}, enzyme mediated cleavage^{23,24}, oxidative stress/ROS²⁵⁻²⁷) or physical (e.g., magnetism²⁸⁻³⁰, temperature³¹, ultrasound^{32,33} or light³⁴⁻³⁶). The latter enable remote activation, converting externally applied energy into localized effects such as heating, mechanical stress, or photon-induced activation at the target site. Depending on the trigger platform used, this may require implantable release reservoirs or local administration of stimulus-responsive carriers (e.g., nanoparticles) that can be subsequently activated *in situ*. As activation relies on energy propagation through biological tissue, achieving precise spatial targeting and accurate dose control remains challenging. In addition, such energy transfer can adversely affect surrounding tissue and, for light-based systems, typically requires optical access to the target region. Iontronic pumps (IP) offer an alternative to several of these limitations by converting electrical signals directly into molecular transport, providing tunable delivery rates with high spatiotemporal control.

Second, we refined the paragraph introducing click-to-release chemistry to distinguish between systemically delivered activators and the localized, electronically controlled approach presented here, and improved the transition toward the selected IEDDA reaction:

This concept has previously been applied for targeted drug activation *in vivo*, including in hydrogel depot systems and antibody-drug conjugates.⁵¹⁻⁵⁴ Although these approaches offer localized therapeutic action, they depend on systemic dosing of prodrugs and afford only limited control over the timing of activation. In contrast, the approach presented here demonstrates electronically controlled activation and enables tunable dosing. To implement this concept, we employed the click-triggered bond-cleavage chemistry of *trans*-cyclooctenes (TCO) and tetrazines (Tz), known for its exceptional kinetics and proven compatibility with biological systems.⁵⁵

Comment #3

I believe the correlation between charge and delivered tetrazine is pretty convincing to me, but the electrochemical efficiency is not fully addressed (I may have missed that part). It would strengthen this section if the authors reported at least an approximate charge-to-molecule conversion ratio. Without that information, it is difficult to assess how much of the applied current contributes to actual ion transport versus leakage. But I understand these may not have been part of the study's design. Even so, the strong linearity between applied charge and delivered amount already supports the claim of precise electrical control. Adding a short statement acknowledging this limitation would make the section feel complete and transparent.

Response:

We revised a statement in the *Results* section quantifying the electrochemical transport efficiency to address the reviewer's comment.

An increase in applied charge (current × time) could be directly correlated to the amount of delivered **1**, demonstrating full control over the iontronic transport (**Error! Reference source not found.c**) with an electrochemical transport efficiency of approximately 68% at +20 nA.

To further clarify the calculation, we added the following description to the *Methods* section:

The electrochemical efficiency was calculated by relating the experimentally delivered amount of Tz **1** to the theoretical amount of **1** defined by the total charge delivered using the following equation:

$$\text{Electrochemical efficiency} = \frac{\text{delivered Tz } \mathbf{1} \text{ (mol)}}{\left(\frac{\text{delivered charge (C)}}{2 \cdot F}\right)} \cdot 100\%$$

Where F is Faraday's constant ($96\,485\text{ C mol}^{-1}$) and the charge number 2 is taken into account to account for the double charge of Tz **1**.

Comment #4:

The linear charge–delivery trend is convincing, yet the error spread in Figure 2c appears uneven and is not discussed. Authors may consider specifying n and providing R^2 to clarify how consistent the linear response is across devices.

Response:

We thank the reviewer for this observation. The number of devices ($n = 4-7$) and the coefficient of determination ($R^2 = 0.9861$) have been added to Fig. 2c to demonstrate the consistency of the linear response. The variation in error magnitude across charge values is characteristic of iontronic devices for this “new geometry” proof-of-principle level of development. Error levels observed at intermediate charge regimes represent typical device-to-device variability, whereas the comparatively lower deviations at minimal and maximal charge transfer likely reflect favorable measurement conditions rather than reduced variability. A systematic investigation of the contributing factors is currently ongoing as part of platform development, but falls outside the scope of the present study, which primarily aims to demonstrate proof-of-principle control rather than device optimization.

Comment #5

This section gives us good chemical evidence through HPLC that the CA4 prodrug can be efficiently cleaved in solution. However, it seems to me this analysis reflects bulk reaction conditions (please correct me if I am mistaken), while the later cell-based results, for example Figure 3c, only infer successful release indirectly through cytotoxicity. As a personal question, don't authors think that, without chemical verification, it remains not more than an assumption that the same release mechanism occurs in the iontronic setup? Including or referencing a chemical confirmation of CA4 formation under iontronic operation would bridge these two pieces of evidence and make the mechanistic link between electrical input and molecular release more realistic.

Response

We agree with the reviewer that the initial HPLC analysis reflected bulk reaction conditions and that direct verification under iontronic operation would strengthen the mechanistic link. To directly confirm that the **CA4** release mechanism occurs under iontronic operation, we performed additional HPLC experiments in which Tz **1** was delivered iontronically into a reservoir containing prodrug **4** ($50\ \mu\text{M}$) for

3, 6, and 9 h. A linear increase in **CA4** formation was observed as a function of pumping duration. In contrast, iontronic delivery of K^+ (0.1 M KCl) for 9 h resulted in no detectable release of **CA4**, confirming both the electrochemical stability of the prodrug and the requirement of Tz transport for activation. These data have been added to the *Supplementary Information 6.1*.

Furthermore following sentence was added to the *Results* section:

To confirm that release also occurs under iontronic conditions, we performed HPLC analysis during electronic delivery of Tz **1** into a reservoir of **4**. **CA4** release increased proportionally with pumping duration (3-9 h), whereas K^+ delivery did not induce any detectable release, confirming mechanistic fidelity and prodrug stability under iontronic operation (see *Supplementary Information 6.1*).

The two **CA4** peaks visible in the HPLC traces correspond to the known E/Z isomers of combretastatin A-4, as routinely observed in analytical separations and reported in the literature (e.g., Karatoprak, G. Ş., Küpeli Akkol (2020). Combretastatins: An Overview of Structure, Probable Mechanisms of Action and Potential Applications. *Molecules*, 25(11), 2560.), <https://doi.org/10.3390/molecules25112560>. This is an expected characteristic of the drug and not indicative of degradation.

Comment #6

Here authors performed an experiment that uses a fixed 4-hour stimulation period at ± 20 nA. This may show on/off functionality but the degree of dose programmability kind of feels uncertain. However, it would help if the authors explained whether shorter activation times or lower currents were tested and yielded some certain effects.

Response

We thank the reviewer for this comment. As detailed in our response to Comment #5, we performed additional iontronic delivery experiments demonstrating that **CA4** release scales proportionally with pumping duration (3-9 h), confirming dose programmability under iontronic control. This data have been included as mentioned in *Supplementary Information 6.1* and the *Results* section.

We would like to emphasize that because **CA4** is highly potent, even short stimulation periods deliver biologically relevant doses, as stated in the manuscript: "Based on the established delivery rate of **1** at

+20 nA, we estimated that active pumping for just 30 seconds would suffice to deliver ~10 nM of Tz into 250 μ L, thereby releasing a biologically relevant amount of CA4.” For this reason, the experiment was designed to demonstrate safe and reversible on/off control rather than fine dosing, which is addressed analytically and mechanistically in the new HPLC data.

Comment #7

In Figure 5b, we can see a background release of BSA and this is shown in red in the figure. Authors wrote: “When KCl solution was pumped instead of tetrazine, a substantial release of BSA was still detected, which we attribute to an electrochemically induced bead detachment effect.” Could you please expand this discussion a bit. I think this part is critical.

Response

We thank the reviewer for this valuable comment. To clarify the origin of the background signal, we emphasize that the click-cleavable linker is chemically stable under iontronic conditions. As demonstrated in the additional HPLC experiment provided in response to **Comment #5** (Supplementary Fig. S10), pumping K^+ does not induce any release from prodrug **4**. Because the BSA conjugate employs the same cTCO linker architecture, we infer equivalent chemical stability in this system.

Given the absence of chemically triggered release under K^+ pumping, we attribute the background observed in the BSA experiment to non-specific effects associated with the bead-based assay format, e.g. passive protein dissociation or electrically induced detachment from the bead surface. To present the iontronically triggered component more clearly and avoid possible misinterpretation, we corrected the data in Fig. 5b in the Results section for background due to non-specific release, while the underlying raw data was added to the Supplementary Information in Section 8.3 for full transparency.

This clarification does not alter the conclusions of the study. It ensures that the electronically triggered release resulting from tetrazine delivery is accurately represented. In subsequent iterations of the platform, we have begun to explore hydrogel-based depot architectures, in which such non-specific effects are anticipated to be attenuated.

Accordingly, the following sentence was revised in the *Results* section to better reflect this explanation:

When delivering K^+ (0.1 M KCl) as a control, we observed a background level of BSA release, which we attributed to passive protein dissociation or electrically induced detachment from the bead surface. This background was subtracted to isolate the iontronic click-to-release component, and the corresponding data are provided in Supplementary Information 8.3. These assay-related effects arise from the bead-based platform and are anticipated to be attenuated in future implementations using hydrogel-based depot architectures.

Reviewer #2

General Comment

The manuscript co-lead by Waldherr, Mikula, and Bintinger demonstrates proof-of-concept for ionotronic click-to-release chemistry of prodrugs and proteins. I enjoyed reading this manuscript, which spans the fields of electroceuticals, bioorthogonal and prodrug chemistry. The manuscript clearly describes precise spatiotemporal control using a model prodrug and masked protein (BSA), thus, will be of great interest to the chemical biology and medicinal chemistry fields. It is also pitched to a level that is accessible for other scientists, especially those in the electrochemical and materials space.

Response

We thank Reviewer #2 for their positive assessment and appreciation of our interdisciplinary approach combining iontronics and bioorthogonal chemistry. We are pleased that the clarity and accessibility of the manuscript were well received.

Comment #1

Something that is not considered here is the lack of regeneration of the depot's payload. Essentially, the TCO is consumed with each reaction and eventually would be exhausted from the depot, rendering it ineffective to further tetrazine released from the ionotronic pump. However, this is another challenge, and it will be exciting to see the next generation of this system and its application to in vivo implants and prodrug activation strategies.

Response

We agree with the reviewer that the current system is designed for single-use activation of immobilized payloads. The present work was intended as a proof-of-concept demonstration, using the bead system as a simplified model platform to establish electronically controlled click-to-release chemistry and to confirm precise, programmable activation of both small and macromolecular payloads.

In the context of highly potent compounds such as **CA4**, regeneration becomes less critical, since realistic depot architectures could serve as effectively inexhaustible reservoirs. Based on feasibility estimations, drug-polymer constructs with approximately 1% (w/w) loading can reach up to 100 μM releasable drug concentrations, several orders of magnitude above typical IC_{50} values for cytotoxins (1–200 nM). Such capacities would enable many activation cycles without significant depletion, even without re-functionalization of the reactive sites. From preliminary *in vivo* studies, administering a sustained local chemotherapy dosing at the tumor site to flank tumors in mice, we observed that the drug dosing can be reduced to only 5% of the systemically efficient dose, still reaching efficient tumor control.

This concept, however, extends beyond the scope of the present study, which focused on establishing feasibility and mechanism using the bead system as a model for future hydrogel-based depots.

Comment #2

The NMR spectra and reported data are good and of very high purity, but there are some NMR assignments that should be corrected (details below).

- Compound 2 has substantially more ^{13}C peaks assigned than is expected. Are these due to rotamers? Also, in the spectra for Compound 2, there are 7 carbon peaks in the 130–185 ppm region (page 28, ESI) but from the structure, I would expect 6 carbon peaks (4 \times carbonyl and 2 \times alkene).
- Compound S3 has 44 assigned carbon peaks, but only 42 expected. Looking at the spectra it appears there are 42 peaks (page 30), so the authors must have assigned extra peaks.
- Compound 2 and S3 have carbon assignments that are identical (e.g., 70.6, 70.6, 70.3 \times 3, etc.). These peaks that are identical should be assigned to 2 decimal places, to distinguish them from each other.
- It is difficult to see using the spectra provided, but some of the coupling constants (J values) for compounds are very large and may not be assigned correctly. Compound 2 is an example where J values of 33 Hz and 22 Hz are reported. Are these correct? Should these peaks that are very complex be assigned as multiplets (m)?

Response

We thank the reviewer for the careful inspection of the NMR data. The following corrections and clarifications were implemented in the *Supporting Information*:

- For Compound 2, we confirmed that the apparent excess ^{13}C peaks result from rotameric conformations, which produce duplicate carbonyl signals. This observation is consistent with previous reports for similar carbamate-containing structures, where Compound 2 has been reported as a mixture of rotamers. The following explanatory sentence was added to the caption: **Compound 2 is reported as a mixture of rotamers** and identical peaks were assigned 2 decimal places to distinguish the signals. The ^{13}C -NMR code reads now:

^{13}C NMR (151 MHz, CD_2Cl_2) δ 160.73, 160.70, 156.0, 155.9, 155.59, 155.57, 154.56, 154.53, 154.49, 154.4, 154.32, 154.29, 154.1, 154.0, 153.9, 152.57, 152.56, 132.1, 132.0, 132.0, 131.9, 125.68, 125.65, 125.62, 125.56, 118.7, 118.52, 118.45, 118.4, 117.8, 117.7, 117.6, 114.5, 114.42, 114.40, 114.37, 110.7, 110.5, 110.4, 75.0, 74.9, 74.8, 74.7, 48.0, 47.8, 47.5, 47.4, 47.3, 47.2, 46.7, 46.5, 41.2, 41.1, 41.0, 36.39, 36.36, 36.21, 36.18, 36.16, 35.8, 35.64, 35.62, 35.5, 35.4, 35.2, 35.0, 34.6, 29.5, 29.4, 24.8, 24.7, 24.5, 18.9.

- For Compound S3, two wrongly assigned carbon signals were removed, and identical peaks were assigned 2 decimal places to distinguish the signals. The ^{13}C -NMR code reads now:

^{13}C NMR (151 MHz, CD_2Cl_2) δ 180.7, 173.2, 163.7, 157.1, 156.1, 132.0, 131.8, 99.1, 72.5, 70.8, 70.8, 70.6, 70.5, 70.40, 70.38, 70.2, 70.1, 63.0, 62.1, 60.5, 55.9, 54.2, 54.0, 53.8, 53.7, 53.5, 46.1, 44.6, 41.3, 41.2, 41.0, 39.7, 39.6, 36.30, 36.29, 36.1, 31.6, 31.4, 29.5, 28.5, 28.4, 26.0, 21.7, 20.5, 18.2, 18.1.

- For Compound 2, overly large J values were re-evaluated. The relevant resonances are indeed complex multiplets, and their coupling constants have been updated or reported as multiplets (m) where appropriate. The ^1H -NMR code reads now:

^1H NMR (600 MHz, CD_2Cl_2) δ 7.59 (ddd, J = 11.8, 8.5, 2.8 Hz, 1H), 7.13 – 7.05 (m, 2H), 6.19 (s, 1H), 5.86 – 5.70 (m, 1H), 5.53 (t, J = 15.5 Hz, 1H), 3.69 – 3.56 (m, 2H), 3.55 – 3.41 (m, 3H), 3.10 (d, J = 10.7 Hz, 2H), 3.01 (dd, J = 12.7, 6.1 Hz, 4H), 2.94 (d, J = 7.6 Hz, 2H), 2.11 – 1.88 (m, 4H), 1.83 (tt, J = 14.4, 7.2 Hz, 1H), 1.73 – 1.54 (m, 3H), 1.51 – 1.42 (m, 1H), 1.13 – 0.99 (m, 1H), 0.85 – 0.73 (m, 1H).

Reviewer #3

General Comment

The authors demonstrated an iontronic system that deliver charged tetrazines as molecular scissors to selectively click trans-cyclooctane linkers and release tethered molecules on demand. They showed that electronically tunable tetrazine delivery sustained over several days and precise release of payloads ranging from small molecules to large proteins. By integrating biorthogonal click-to-release chemistry with iontronic, the current work overcomes molecular size, charge and stability limitations.

Strengths of the manuscript:

(i) Click-to-release is an interesting concept that beautifully utilizes charged tetrazines as molecular scissors to cleave the bond.

(ii) This concept overcomes the limitation of delivering large molecular weight payload.

(iii) The molecular conjugates were thoughtfully designed and characterized. The experimental demonstrations are clear in most cases.

Response:

We thank Reviewer #3 for the positive and encouraging evaluation. We appreciate the recognition of the novelty and clarity of our approach, as well as the acknowledgment of the system's ability to achieve electronically controlled, on-demand release of both small molecules and proteins. The comments accurately capture the conceptual advance of coupling iontronic delivery with bioorthogonal click-to-release chemistry, and we are pleased that the strength and design of the molecular conjugates were well received.

Weaknesses of the manuscript (i/ii)

Comment (i)

The concept is solely dependent on molecular design. The author has shown only one transporter molecule, i.e., tetrazine.

Response

The focus of the present study was not to establish a general mechanism of iontronic transport, which has already been demonstrated in the literature for a variety of drug molecules already mentioned in the Introduction of the current paper, but rather to show that click-to-release chemistry can be integrated with iontronic delivery to enable remotely controlled activation. We therefore selected one representative charged tetrazine species that allows precise electrical addressing and mechanistic clarity. This approach ensured that the study could clearly demonstrate the feasibility of electrically triggered bioorthogonal release. We fully agree that extending this principle to other click-to-release triggers will further broaden the chemical scope of this activation strategy, and this is the direction of our current work. While not shown in this manuscript, our preliminary experiments with additional charged tetrazine triggers, including benzylic amine 4-(6-methyl-1,2,4,5-tetrazin-3-yl)phenyl)methanamine, as well as the unsymmetrical variant of **1** - 2-(6-methyl-1,2,4,5-tetrazin-3-yl)ethan-1-amine (see. Sarris, *Chem. Eur. J.* **2018**, *24*, 18075, <https://doi.org/10.1002/chem.201803839>), indicate comparable iontronic transport and click-to-release performance. This supports that the concept is not limited to a single tetrazine structure, and a systematic evaluation of this broader tetrazine library is currently ongoing.

Comment (ii)

Solubility of the conjugate molecule might be a major concern.

Response

We appreciate this point and confirm that all compounds used in this study were designed to exhibit adequate aqueous solubility under the experimental conditions. The Tz **1** carries two permanent charges, ensuring good solubility and efficient iontronic transport. The sulfonated CA4 conjugate **4** and its released product **CA4** were proven to be soluble at the concentrations applied in both chemical and biological assays (see, e.g., Keppel, *ChemBioChem* 2022, 23, e202200363. <https://doi.org/10.1002/cbic.202200363>), and BSA is known to remain fully soluble and structurally stable under similar buffered conditions (see, e.g., Peters, *Adv. Protein Chem.* 1985, 37, 161–245. [https://doi.org/10.1016/S0065-3233\(08\)60065-0](https://doi.org/10.1016/S0065-3233(08)60065-0)).

We also note that solubility is not expected to be a limiting factor in future advanced depot structures, such as hydrogels, where the conjugates are immobilized within a solid support. In such systems, only the solubility of the released payloads will be relevant. Here the electrical control over release rate allows precise adjustment to prevent local precipitation even for less soluble payloads.

Comment #1

The release kinetics of the payload are likely governed by the formulation of the cation-exchange membrane (PEGDA/AMPSA), since crosslink density and membrane thickness strongly influence diffusion and how long tetrazine is retained in the matrix. Did the authors evaluate these effects (PEGDA:AMPSA ratio, degree of polymerization/crosslinking, and film thickness)? A plot showing component concentration (or PEGDA:AMPSA ratio / crosslink density) versus tetrazine retention time (with membrane thickness as a separate series or secondary axis) may clarify how membrane composition can control release kinetics.

Response

We fully agree that the membrane composition and thickness critically affect ion transport and retention. In the present work, our goal was to demonstrate the feasibility and electronic control of tetrazine delivery, and therefore we employed a standardized PEGDA/AMPSA formulation optimized in previous iontronic studies to ensure reproducible transport characteristics. For previous studies see:

Correlating Ionic Conductivity and Microstructure in Polyelectrolyte Hydrogels for Bioelectronic Devices - *Macromol. Rapid Commun* (2022).
<https://doi.org/10.1002/marc.202100687>

Translating Electronic Currents to Precise Acetylcholine-Induced Neuronal Signaling using an Organic Electrophoretic Delivery Device - *Advanced Materials* (2009).
<https://doi.org/10.1002/adma.200900187>

Capillary-fiber based electrophoretic delivery device - *ACS Applied Materials & Interfaces* (2019).
<https://doi.org/10.1021/acsami.8b22680>

A decade of iontronic delivery devices - *Advanced Materials Technologies* (2018).
<https://doi.org/10.1002/admt.201700360>

Impact of PEGMA on transport and co-transport of methanol and acetate in PEGDA-AMPS cation exchange membranes - *Journal of Membrane Science* (2022).
<https://doi.org/10.1016/j.memsci.2021.119950>

To avoid introducing additional variables that would obscure the relationship between applied charge and tetrazine dose, we deliberately kept PEGDA:AMPSA ratio, crosslinking conditions, and membrane thickness constant.

A systematic investigation of how membrane composition, crosslinking degree, and thickness influence release kinetics is indeed important and represents the next step toward application-specific optimization. However, this extends beyond the scope of the current proof-of-concept study, which focused on coupling iontronic delivery with chemical activation.

Comment #2

The rationale for using beads is not clearly explained in the manuscript. Could the authors clarify the choice? Does the size of the beads influence the outcome?

Response

Streptavidin-coated magnetic beads were selected as a well-defined, commercially available model platform that allows highly reproducible immobilization of biotin-tagged conjugates via a well-established biotin-streptavidin interaction. Their surface-based presentation of cTCO-functionalised molecules enables precise control over the number of reactive sites and facilitates quantitative release analysis. While bead size may, in principle, affect total surface area, all experiments were carried out using the same batch of beads under identical conditions, ensuring internal comparability.

The bead setup was therefore intended as a rapid and analytically accessible proof-of-concept system to demonstrate iontronic control over chemical release. It does not represent a final depot design for *in vivo* use. Ongoing work focuses on advancing the immobilization strategy toward hydrogel-based depot architectures, which are more suitable for biological environments and scalable therapeutic implementation.

To clarify this in the manuscript, we updated the sentence in the *Results* section as follows:

Building on these results, we employed cTCO-bis-NHS (5) to synthesize biotin-cTCO-DMEDA-CA4 (6), enabling immobilization of the prodrug onto streptavidin-coated magnetic beads, which were selected as a well-defined solid support allowing quantitative analysis of release while serving as a simplified proof-of-principle model system.

Comment #3

In Figure 2e, for a constant current of +5 nA, the delivery rate was 0.7 ± 0.6 pmol min⁻¹. This value exhibits a very large relative error. Comment on whether the applied current was too insignificant to trigger any mobility.

Response

We would like to clarify that iontronic transport was clearly observed at +5 nA, confirming that the applied current was sufficient to drive tetrazine delivery. The larger relative error at this low current stems from the inherent variability of the first-generation devices, as indicated by the comparable absolute standard deviation across all current levels. While the relative variation appears larger at +5

nA due to the smaller delivered amount, the underlying transport mechanism remains fully active and scales linearly with the applied charge.

We are currently working on further optimization of both the device architecture and analytical assays, aiming to improve precision and reproducibility in future generations of iontronic delivery systems. The overall linear correlation between applied charge and delivered amount remains consistent, confirming robust electrical control even in this initial prototype configuration.

Comment #4

The second-order rate constant ($46 \text{ M}^{-1}\cdot\text{s}^{-1}$) was determined using a model cTCO compound. How well do you expect this kinetic value to translate to the actual prodrug (compound 4) in the cellular environment, where diffusion and steric factors differ?

Response

We agree that the cellular environment can influence the apparent click-to-release kinetics. To address this directly, we determined the kinetics in a representative cell-culture medium (Fluorobrite supplemented with 10% FBS), which yielded a **second-order rate constant of $37 \text{ M}^{-1}\cdot\text{s}^{-1}$** , confirming that the reaction proceeds efficiently under physiologically relevant conditions.

The experiment was added to the *Supporting Information* (Supplementary Fig. S1) and the second-order rate constant was adapted in the *Results* section:

Kinetic data with exponential fits

37 °C, DMSO content: <1%

Fig. S1 Kinetic data (pink curves, six measurements) with exponential fits (dashed lines) of Tz 1 reacting with cTCO(PEG₂-OH)₂ (3) in aqueous media at 37 °C.

Determining rate constants with a model TCO is a standard practice to benchmark intrinsic reactivity for related conjugates that share the same reactive core - e.g.:

Unraveling Tetrazine-Triggered Bioorthogonal Elimination Enables Chemical Tools for Ultrafast Release and Universal Cleavage - *Journal of the American Chemical Society* (2018).
<https://doi.org/10.1021/jacs.7b11217>

The overall click-to-release process is governed primarily by the used Tz-TCO combinations with limited influence from the nature of the leaving group/payload, supporting the use of model systems for rate benchmarking.

Beyond the current study, we are developing next generation click-to-release linkers based on the recently reported iTCO scaffold, which shifts control of the post click cascade to the TCO and enables ultrafast, near quantitative release across a broad range of tetrazines and biological media. Integrating

iTCO into our iontronic platform is expected to further increase release efficiency under physiologically relevant conditions, but this lies beyond the scope of the present work.

Pre-print: <https://chemrxiv.org/engage/chemrxiv/article-details/67ce0a786dde43c90833b680>

Comment #5

Did you examine the stability of the prodrug in cell culture medium or serum over time (without Tz), to confirm no slow, non-specific release of CA4 occurs?

Response

The stability of the identical prodrug (compound 12, sulfo-cTCO-DMEDA-CA4) has been comprehensively characterized in our previous work (*ChemBioChem*, 2022 – Reference 42 in the original manuscript). HPLC analysis confirmed that the compound remained fully stable in both PBS and cell growth medium (DMEM + 10 % FBS) at 37 °C for 120 h, showing no detectable release of CA4. Only partial *trans*-to-*cis* isomerization of the cTCO linker ($\approx 33\%$ after 72 h, $\approx 44\%$ after 120 h) was observed, while the DMEDA bis(carbamate) linkage remained intact, confirming the integrity of the prodrug under physiological conditions.

These findings verify that the CA4-TCO prodrug **4** used in this study is chemically stable and that drug release occurs exclusively upon tetrazine activation. Cell-viability data for **4** in Fig. 3b in the current manuscript further confirms this assumption.

Please note that, such *trans* to *cis* isomerization would render the click-to-release pathway ineffective for that fraction of molecules, but it would merely keep **CA4** covalently caged rather than generate free drug. From a safety perspective, this means that no unintended payload is released and that isomerization can only reduce the fraction of prodrug that remains addressable, not create background toxicity.

Tetrazine-Triggered Bioorthogonal Cleavage of trans-Cyclooctene-Caged Phenols Using a Minimal Self-Immolative Linker - *ChemBioChem* (2022). <https://doi.org/10.1002/cbic.202200363>

Comment #6

The covalent attachment of biotin-cTCO to BSA via SPAAC is clear. However, it would be valuable to report the average number of cTCO groups per BSA molecule (degree of functionalization), and additionally any potential protein denaturation during SPAAC attachment or immobilization could affect release and activity should be addressed.

Response

The BSA-azide used in this study is a commercial mono-functionalized derivative (modified at the single accessible cysteine residue), which ensures a defined 1:1 correlation between the reactive site and the released protein. We added the following clarification in the manuscript:

To verify that iontronic click-to-release can be applied to payload delivery independent of molecular weight, we demonstrated release of immobilized BSA (66 kDa) using the identical bead-based model system (**Error! Reference source not found.a**). Therefore, a bicyclo[6.1.0]non-4-yne (BCN) moiety was installed analogously to the synthesis of **6**, which enabled the covalent attachment of commercial mono-functionalized BSA-azide (**BSA-N₃**) to afford biotin-cTCO-BSA (**7**) via strain-promoted azide-alkyne cycloaddition (SPAAC). The mono-functionalization of BSA ensured site-specific conjugation

and a defined 1:1 stoichiometry between protein and cTCO. Regarding denaturation: SPAAC is widely applied under mild, aqueous conditions for protein bioconjugation (including *in vivo* labeling), and streptavidin-biotin magnetic bead systems-specifically Dynabeads™ MyOne™ Streptavidin C1 are routinely used for protein capture/immobilization in immunoassays and pull-downs under non-denaturing conditions. Based on this established practice and manufacturer guidance, we do not expect denaturation under our experimental conditions – see:

Dynabeads™ MyOne™ Streptavidin C1 – product protocol page (Thermo Fisher):

<https://www.thermofisher.com/us/en/home/references/protocols/proteins-expression-isolation-and-analysis/protein-isolation-protocol/dynabeads-myone-streptavidin-c1.html>

Example of protein immobilization on streptavidin surfaces under non-denaturing assay conditions

(supports common use in ELISA/pull-downs): *PLOS ONE* (2018): “Biotin-tagged proteins: Reagents for efficient ELISA-based antibody analysis”

<https://journals.plos.org/plosone/article?id=10.1371%2Fjournal.pone.0191315>